# Space-Correlated Transformer: Jointly Explore the Matching and Motion Clues in 3D Single Object Tracking

## Abstract

3D Single Object Tracking (3D SOT) in LiDAR point clouds plays a crucial role in autonomous driving. Current approaches mostly follow two paradigms, i.e., Siamese matching-based and motion-centric. However, LiDAR point clouds lack enough appearance information, while the motion-centric trackers suffer from complex model structures. To address these issues, we present a novel and conceptually simple tracking framework dubbed SCtrack, which jointly explores the matching and motion clues in point clouds. Specifically, SCtrack embeds point clouds into spatially structured features and conducts space correlation along the aligned spatial region. The target relative motion is directly inferred from the correlated features. In contrast to prevalent PointNet-based features, our spatially structured representation inherently models motion clues among the consecutive frames of point clouds, thereby being complementary to appearance matching. To better utilize the aligned structured features, we employ a strategy of varied-size space regions that adapt to different target shapes and locations during space correlation. Without bells and whistles, SCtrack achieves leading performance, with 89.1%, 71.5%, and 62.7% precision on KITTI, NuScenes, and Waymo Open Dataset, and runs at a considerably high speed of 60 Fps on a single RTX3090 GPU. Extensive studies validate the effectiveness of our SCtrack framework. The code will be released.

## 1 Introduction

3D single object tracking (SOT) based on point clouds is a fundamental task with enormous potential for various applications, including autonomous driving and robotics (Lee & Hwang, 2015; Liu et al., 2018; Wu et al., 2022; Zhu et al., 2022). Early efforts (Kristan et al., 2015; 2016; Bertinetto et al., 2016b; Danelljan et al., 2017; Li et al., 2018; Xie et al., 2022; Ye et al., 2022) focus on visual object tracking that uses RGB images obtained by cameras. Recently, with the development of 3D sensors, such as LiDAR, 3D data is easy to acquire and set up for 3D object tracking. However, due to the sparsity and irregular distribution of 3D points, existing popular schemes on 2D visual tracking cannot be directly applied to 3D single object tracking. As a result, accurately and efficiently analyzing point cloud data to track objects in complex scenes is still a challenging and open problem.

Many current tracking approaches (Wang et al., 2021c; Cui et al., 2021; Zheng et al., 2021; Shan et al., 2021; Hui et al., 2021; 2022) use a point-wise representation by taking raw point clouds as input. For instance, P2B (Qi et al., 2020) and its subsequent works use a point-based network with a Siamese architecture for feature extraction. This is followed by a point-wise appearance-matching module that propagates target cues and a target prediction network, as shown in Fig. 1 (a). M2-Track (Zheng et al., 2022) proposes a motion-centric approach that first identifies the target points by segmenting them from their surroundings using a PointNet (Qi et al., 2017a) segmentation network. Then, it leverages motion clues to refine the localization of the target, as shown in Fig. 1 (b). Although these approaches (Giancola et al., 2019; Qi et al., 2020; Fang et al., 2020; Wang et al., 2021c; Cui et al., 2021; Zheng et al., 2021; Shan et al., 2021; Hui et al., 2021; 2022) have shown excellent performance in tracking benchmarks, motion-centric trackers often necessitate complex model structures. In the meantime, Siamese matching trackers suffer from insufficient appearance

information in point-wise features. Therefore, a natural question arises: can we simultaneously leverage both motion and appearance clues using a conceptually simple model?

In this paper, we present a novel and conceptually simple tracking paradigm, dubbed SCtrack, which jointly explores appearance matching and motion clues for 3D SOT, as shown in Fig. 1 (c). To ensure efficiency and simplicity, we combine the separated feature extraction and matching in the conventional Siamese paradigm and remove the two stages of the motion modeling paradigm with one single network. Our proposed backbone network consists of two core parts: the structured embedding module and the Space-Correlated Transformer (SCT) module. In contrast to the widely adopted point-wise feature networks, e.g., PointNet (Qi et al., 2017a) and PointNet++ (Qi et al., 2017b), our structured embedding module adopts spatially aligned structure representations for better appearance matching and motion modeling. Specifically, to exploit the spatial relations among tracked targets and distractors, we subdivide the 3D space into equally spaced voxels given a pair of point clouds from the template and search regions. With structured voxel features, we squeeze, concatenate, and merge the template and search features along the channel dimension. Then, a space-correlated

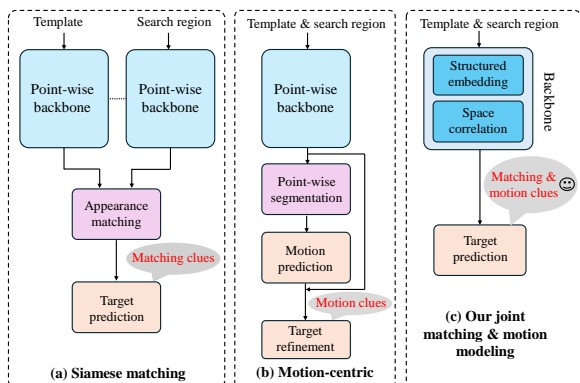

Figure 1: Comparison with typical 3D SOT paradigms. (a) Siamese matching paradigm (Giancola et al., 2019; Qi et al., 2020) adopts point-wise representation (Qi et al., 2017b) and conducts appearance-matching operations, neglecting point clouds' insufficient appearance and shape information. (b) motion-centric (Zheng et al., 2022) leverage motion modeling while suffering complex two-stage model structure. (c) On the contrary, our proposed SCtrack paradigm jointly explores appearance and matching information, greatly retaining the spatial information and remaining a conceptually simple tracking pipeline.

transformer is proposed to exploit the motion and appearance clues on the structured features. SCT partitions the merged template and search features into different space regions for correlation. Intuitively, using a fixed-size region may be sub-optimal for dealing with targets of different sizes. To better adapt to the diverse shapes and motion patterns of the tracked targets, we use a tiny hyper-network module to predict the size and location of the correlated space regions. The hyper-network module can learn the diverse correlated region size depending on the template and search region features as inputs. Thus, the rich motion and appearance clues among two consecutive frames can be well-captured, while the feature correlation can also fulfill the appearance matching. In the final, we can leverage a single backbone network, which includes the structured embedding module and space-correlated transformer, to realize synchronized matching and motion modeling.

In conclusion, the contribution of this study is threefold as follows:

- We present a joint matching and motion modeling tracking paradigm, which overcomes the inability of the traditional Siamese paradigm and is conceptually simpler than the motion-centric paradigm in the model pipeline.
- We adopt spatially structured representation and propose a novel space-correlated transformer with adaptive region size, greatly advancing the feature learning of 3D single object tracking in point clouds.
- Our joint matching and motion modeling method, SCtrack, achieves state-of-the-art performance on the KITTI, nuScenes, and Waymo datasets in 3D single object tracking while running at a considerably fast speed.

## 2 RELATED WORK

**2D Single Object Tracking.** 2D Single object tracking in videos is closely related to 3D point cloud tracking tasks. The dominant 2D object tracking paradigm is Siamese tracking (Bertinetto et al.,

2016a; Li et al., 2018; Zhang & Peng, 2019; Xu et al., 2020), which regards the tracking problem as an appearance-matching problem between consecutive video images. Due to the rich appearance clues and complex motion patterns in natural 2D video data, most 2D trackers (Chen et al., 2021; Yan et al., 2021; Cui et al., 2022; Xie et al., 2022; Ye et al., 2022) focus on enhancing appearance matching. The representative trackers are based on transformers (Vaswani et al., 2017; Dosovitskiy et al., 2021). For example, TransT (Chen et al., 2021) conducts the cross-attention (Vaswani et al., 2017) module multiple times to improve the matching quality. Stark (Yan et al., 2021) manages a dynamic template to obtain a better appearance-matching result over time. Despite their great success, the 2D tracking techniques largely overlook the motion clues, hindering their direct application in 3D point cloud tracking. Realizing that point cloud data lacks appearance information, we jointly model the motion and appearance clues instead of solely exploiting one clue.

**3D Single Object Tracking.** 3D Single Object Tracking shares a similar task definition with 2D object tracking. Consequently, classical 2D tracking techniques such as Siamese network (Bertinetto et al., 2016a) and region proposal networks (Ren et al., 2015; Li et al., 2018), are widely transferred to the 3D point cloud tracking domain. The pioneering 3D SOT work, P2B (Qi et al., 2020) follows the Siamese paradigm and develops a Point-to-Box network for target box regression. With a simple Siamese tracking pipeline, most 3D SOT methods (Qi et al., 2020; Hui et al., 2021; Xu et al., 2023b) focus on enhancing target-matching modules and prediction networks. Among them, the transformer-based methods (Hui et al., 2022; Zheng et al., 2021; Shan et al., 2021) are most widely explored to enhance the point cloud matching for tracking. For example, PTTR (Zhou et al., 2021) proposes a Point Relation Transformer to match the appearance clues in point clouds of the template and search region. Later, CXTtrack (Xu et al., 2023a) conducts transformer matching in both local and global contexts to exploit the appearance clues more comprehensively. MBPtrack (Xu et al., 2023b) further improves the appearance-matching paradigm by using a memory network to leverage the historical point clouds. SyncTrack (Ma et al., 2023) performs synchronizing feature extraction and matching inspired by the one-stream pipeline (Ye et al., 2022). Thus, most 3D methods are developed under the Siamese appearance-matching paradigm. Differently, M2Track (Zheng et al., 2022) firstly builds a pure motion-centric paradigm rather than exploiting appearance clues solely. M2Track models the motion clues by processing two consecutive frames in two stages: A target segmentation to classify the foreground object and a target prediction refinement to localize the target precisely.

Despite their great success, most 3D trackers improve tracking performance by developing more complex appearance-matching modules because they overlook the role motion clues. The point cloud data modality is generally textureless, unordered, and sparse, which makes appearance-matching less effective than that of 2D object tracking in videos. However, the motion-centric tracker M2Track suffers from a complex two-stage model pipeline. In this work, we joint exploit appearance matching and motion modeling by constructing a conceptually simple framework, which fully leverages spatially structured feature representation to guide accurate target localization.

**Transformer in 2D/3D Vision.** Transformers have shown great performance and wild applications in various vision tasks. The attention scheme (Vaswani et al., 2017) exhibits a stronger global modeling ability than the traditional CNN (Krizhevsky et al., 2012; He et al., 2016) architecture in the visual domain. ViT (Dosovitskiy et al., 2021) first conducts a comprehensive exploration of the transformer-based backbone network for the fundamental image classification task. Then, vision transformers are modified and adapted to various 2D vision tasks. SwinT (Liu et al., 2021) and PVT (Wang et al., 2021a) adopt the hierarchical structure with an approximation attention scheme for dense visual prediction tasks, such as object detection (Lin et al., 2014) and segmenation (Wang et al., 2021b). In the 3D vision task, the transformer also shows superiority in modeling the point cloud data. The point transformer (Zhao et al., 2021) is the pioneering work to apply transformer in 3D point cloud understanding task. Later, the succeeding works apply a transformer to more widely 3D tasks, such as 3DETR (Misra et al., 2021) for detection, Segformer3D (Perera et al., 2024) for segmentation, and PointformerV3 (Wu et al., 2024) for indoor/outdoor point cloud tasks. Transformer also demonstrates the effectiveness in processing consecutive point clouds, such as 3D video detection and tracking (Yin et al., 2021).

In the 3D single object tracking area, transformer-based trackers (Hui et al., 2022; Zhou et al., 2021; Shan et al., 2021; Nie et al., 2023) particularly address the appearance matching problem, largely neglecting the motion clues in consecutive point clouds. In this work, we first embed the point

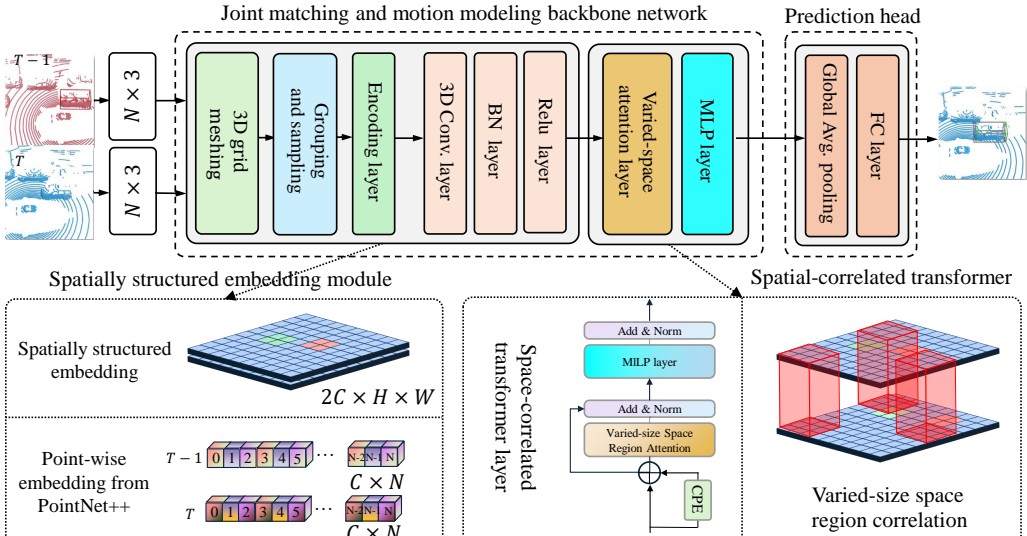

Figure 2: The detailed architecture of our joint matching and motion modeling tracking pipeline. Our simple tracking paradigm comprises a backbone network for joint modeling and a prediction head for directly outputting the relative target motion. The proposed feature network consists of a spatially structured embedding module and a space-correlated transformer. FC layer denotes the fully connected layers.

cloud data into a structured space region and then leverage the transformer's strong ability to exploit motion and appearance clues jointly.

## 3 METHOD

### 3.1 PROBLEM DEFINITION

The 3D Single Object Tracking (SOT) task definition is to find a specific object by giving the target point clouds in the first frame. The target state is generally described by a seven-dimensional 3D bounding box $(x, y, z, w, l, h, \theta) \in \mathbb{R}^7$. Specifically, $\{xyz, wlh, \theta\}$ describes the target center, object size, and orientation angle, respectively. Considering that most objects in point cloud tracking scenarios are rigid, the trackers only need to predict the relative translation $(\Delta x, \Delta y, \Delta z)$ and changing angle $\Delta \theta$. A typical working pipeline of 3D SOT trackers is described as follows: Given the template point cloud $\mathcal{P}_{t=1} \in \mathbb{R}^{N \times 3}$ ($N$ is the number of point clouds) and initial bounding box $\mathcal{B}_1$ in the first frame, the tracker predicts the relative movement for incoming search region point clouds $\mathcal{P}_{t=i} \in \mathbb{R}^{N \times 3}$ ($i$ is the current time index). The final predicted 3D bounding box $\mathcal{B}_i$ for localizing the current target is obtained by a rigid body transformation, which applies the relative movement $(\Delta x, \Delta y, \Delta z)$ to the previous 3D bounding box $\mathcal{B}_{i-1}$.

### 3.2 JOINT MATCHING AND MOTION MODELING PIPELINE

We propose to unify the independent feature extraction and matching in the Siamese paradigm and remove the two stages of the motion-centric paradigm with one single backbone. Within a backbone network, we first use the structured embedding module to process unstructured points. Then, we adopt a space-correlated transformer for feature correlation and motion modeling. Finally, the target position is directly predicted from the extracted features. Our SCtrack can realize synchronized matching and motion modeling.

**Spatially structured embedding.** The proposed Spatially structured network consists of two functional processes: (1) Voxel generation and (2) Voxel feature encoding, as illustrated in Fig. 2 and Fig. 3. We provide a detailed introduction to the spatially structured module as follows:

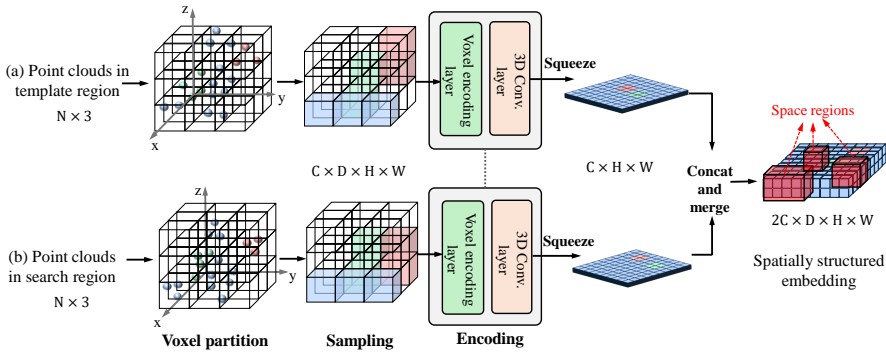

Figure 3: The details of spatially structured embedding module. The structured features of the template and search region are squeezed, concatenated, and merged to capture the target motion clues. In contrast to the popular point-wise features, spatially structured feature representation can preserve more spatial information for predicting relative target motion in two consecutive frames.

To accurately capture the relative motion of the tracked target, we subdivide the 3D space into equally spaced voxels given a pair of point clouds from the template and search regions. Suppose the point cloud encompasses 3D space with range $D$, $H$, $W$ along the $z$, $y$, $x$ axes respectively. We define each voxel of size $v_D$, $v_H$, and $v_W$ accordingly. The resulting 3D voxel grid is of size $D' = D/v_D, H' = H/v_H, W' = W/v_W$. Due to the sparsity and highly variable point density throughout the space, we group the points according to the voxel they are located in. Then, we randomly sample a fixed number, $N$, of points from those voxels containing more than $N$ points. To encode the shape of the surface contained within the voxel, we adopt the hierarchical feature encoding layer in VoxelNet Zhou & Tuzel (2018) to process each voxel. Then, we obtain voxel features $\mathbf{VF}_{\{C,D,H,W\}}$ with the grid size $\{v_D, v_H, v_W\}$ and channel dimension $C$, repectively. We further feed voxel features $\mathbf{VF}_{\{C,D,H,W\}}$ into 3D convolutional layers, which consist of stacked 3D convolution, BN Ioffe & Szegedy (2015) layer, and ReLU Real et al. (2019) layer. The 3D convolutional layers aggregate voxel-wise features within a progressively expanding receptive field, adding more context to the shape description. The detailed architectural variants of the embedding module are in supplementary materials.

*Comparisons with point-wise feature embedding.* The mainstream 3D single object trackers commonly adopt point-wise feature representation extracted by PointNet Qi et al. (2017a;b). However, the PointNet-based features emphasize appearance matching more while neglecting the spatial relationship among points. In contrast, our spatially structured representation inherently reflects the spatial relationships among corresponding points, thereby guiding a more accurate relative motion of the target between two consecutive frames for 3D tracking tasks.

## 3.3 SPACE-CORRELATED TRANSFORMER

Space-Correlated Transformer (SCT) is built on top of the basic structure of the transformer layer and serves as a feature correlation module for the template and search region, as shown in Fig. 3. After obtaining the embedding $\mathbf{VF}_{\{C,D,H,W\}}$, we use SCT to capture the appearance and motion clues within a structured space region (see red cubes in Fig. 3). Rather than using fixed-size space region to conduct spatially structured correlation, our SCT learns to flexibly determine the space region size, i.e., spatially aligned space area, given template/search region structured input.

**Varied-size space generation.** Technically, given the voxel-based structured features $\mathbf{VF}^{\text{temp}}_{\{C,D,H,W\}}$ and $\mathbf{VF}^{\text{search}}_{\{C,D,H,W\}}$, SCT first squeezes structured features along the height dimension and concatenates the template and search region features along the channel dimension. Then, SCT merges structured features along the spatial dimension, i.e., the XOY plane. Thus, we spatially align the structured features of the template and search region, which are for the next-step joint modeling:

$$\mathbf{VF}^{\text{temp}}_{\{C,H,W\}}, \mathbf{VF}^{\text{search}}_{\{C,H,W\}} = \mathbf{Squeeze}(\mathbf{VF}^{\text{temp}}_{\{C,D,H,W\}}, \mathbf{VF}^{\text{search}}_{\{C,D,H,W\}}),$$
$$\mathbf{VF}^{\text{all}}_{\{C+C,H,W\}} = \mathbf{Concat}(\mathbf{VF}^{\text{temp}}_{\{C,H,W\}}, \mathbf{VF}^{\text{search}}_{\{C,H,W\}}), \quad (1)$$
$$\mathbf{VF}^{\text{all}}_{\{2C,H,W\}} = \mathbf{Merge}(\mathbf{VF}^{\text{all}}_{\{C+C,H,W\}}),$$

Figure 4: The details of varied-size space region correlation transformer module, SCT. Varied-size space regions are learned to capture the motion clues of the structured features and adapted to different target sizes.

where $\{\textbf{Squeeze}, \textbf{Concat}, \textbf{Merge}\}$ denotes the squeeze, concatenation and merge operation. Since each feature vector in $\textbf{VF}^{\text{all}}_{\{2C,H,W\}}$ spatially aligns the features of the template and search region, we partition the feature into space regions $\text{VF}_{R|\{1,2,...,M\}}$ with the $M$ predefined regions in XOY plane. We refer to these regions as default regions, and details are in supplementary materials.

To better adapt to the diverse target shapes and motion patterns, it is critical to transform default regions into varied-size regions. As shown in Fig. 4, we adopt two transforming factors, i.e., scale factor $S_r$ to modulate the spatial size and offset factor $O_r$ to translate the default regions in the horizontal and vertical direction. Inspired by the Rao et al. (2021); Meng et al. (2022); Zhang et al. (2022), we use a tiny hyper-network to predict these two transforming factors:

$$S_{r|\{1,2,...,M\}}, O_{r|\{1,2,...,M\}} = \textbf{HyperNetwork}(\textbf{VF}^{\text{default}}_{R|\{1,2,...,M\}}), \tag{2}$$

the hyper-network is highly efficient, which only consists of an average pooling layer, a LeakyReLU (Xu et al., 2015) activation layer, and a $1 \times 1$ fully-connected layer with stride 1. Then, we obtain feature partitions in varied-size regions:

$$\textbf{VF}^{\text{varied}}_{R|\{1,2,...,M\}} = \textbf{Transform}(\textbf{VF}^{\text{default}}_{R|\{1,2,...,M\}}|S_{r|\{1,2,...,M\}}, O_{r|\{1,2,...,M\}}), \tag{3}$$

then, the feature correlation is conducted within the generated regions.

**Space-correlated attention.** Once varied-size space regions are generated, we conduct correlation to capture the motion and appearance clues between template and search regions. We first get the query features from the default regions using a linear projection layer, i.e.,

$$Q_r = \textbf{LinearProject}(\textbf{VF}^{\text{default}}_{R|\{1,2,...,M\}}). \tag{4}$$

As the hyper-network has a restricted receptive field, we use the query features from the default regions and key/value features from the corresponding varied-size regions to build cross-region dependencies. Then, we sample the key and value tokens $\{K_r, V_r\}$ from the learned varied-size regions, i.e.,

$$K_r, V_r = \textbf{LinearProject}(\textbf{VF}^{\text{default}}_{R|\{1,2,...,M\}}|S_{r|\{1,2,...,M\}}, O_{r|\{1,2,...,M\}}). \tag{5}$$

The sampled vectors $K_r, V_r$ are then fed into Multi-Head Self Attention (MHSA) (Dosovitskiy et al., 2020) with queries $Q_r$ for attention computation:

$$Q_r^{\text{attn}} = \textbf{MHSA}(Q_r|(K_r, V_r)). \tag{6}$$

However, as the key/value vectors are sampled from different locations with the query vectors, the relative position embeddings between the query and key vectors may not describe the spatial relationship well. Following (Chu et al., 2021b; Wang et al., 2021a), we further adopt conditional position embedding (Chu et al., 2021a) (CPE) before the MHSA layers, i.e.,

$$\textbf{VF} = \textbf{VF}^{l-1} + \textbf{CPE}(\textbf{VF}^{l-1}), \tag{7}$$

where $\textbf{VF}^{l-1}$ is the structured feature from the previous transformer layer. CPE (Chu et al., 2021b) is implemented by a depth-wise convolution layer. Details can be found in supplementary materials.

*Comparisons with 3D Siamese tracking.* Siamese-based trackers mostly follow the "Extracting then Matching" paradigm and neglect the motion clues in 3D tracking. However, the insufficient

appearance information in point clouds hinders Siamese trackers from effective object tracking. Motion-centric trackers, e.g., M2Track (Zheng et al., 2022) suffer from complex model structures and multi-stage training. In contrast, our joint matching and motion framework achieves a win-win scenario: it allows for effective motion modeling and adopts a pseudo-Siamese structure, which is much simpler than the motion-centric model pipeline.

### 3.4 PREDITION NETWORK AND TRAINING LOSS

**Prediction head.** Perious works (Xu et al., 2023a) generally adopt RPN-style (Ren et al., 2015), multi-branch (Zheng et al., 2022), and transformer-based (Xu et al., 2023b) head network to predict the target scores and corresponding relative motion. Differently, we adopt an extremely simple head and loss design that directly regresses the 4 Degree-Of-Freedom(DOF) relative motion of the target. Specifically, we conduct a max-pooling to the correlated feature $\mathbf{VF}^{\text{all}}_{\{2C,H\times W\}} \in \mathcal{R}^{D\times C}$ and apply a linear layer to obtain the results:

$$\mathcal{RM}_{t-1,t} = \textbf{LinearLayer}\left(\textbf{MaxPooling}\left(\mathbf{VF}^{\text{all}}_{\{2C,H\times W\}}\right)\right), \tag{8}$$

where $\mathcal{RM}_{t-1,t} = (\Delta x_t, \Delta y_t, \Delta z_t, \Delta \theta_t)$ is the relative motion between current frame $t$ and previous frame $t-1$. Since we do not adopt an ensembling prediction like RPN, the current target 3D bounding box can be directly inferred from the previous box position $\mathcal{B}_{t-1}$ through a simple rigid transformation (Hui et al., 2022):

$$\mathcal{B}_t := (\mathcal{B}_{t-1} \xrightarrow{\mathcal{RM}_{t-1,t}} \mathcal{B}_t), \tag{9}$$

where a rigid body transformation $\xrightarrow{\mathcal{RM}_{t-1,t}}$ transfers the relative movement result into the current target position per frame. Thus, the inference pipeline is also simplified as the ensembling predictions are replaced by direct regression.

**Training loss.** Due to the simplified design of the prediction head, our SCtrack model is trained to regress the 4-DOF relative motion state $(\Delta x_t, \Delta y_t, \Delta z_t, \Delta \theta_t)$ directly. A simple Maximum Likelihood Estimation (MLE) (Li et al., 2021) loss is adopted to supervise the whole model pipeline. More details can be found in supplementary materials.

## 4 EXPERIMENTS

### 4.1 EXPERIMENTAL SETTINGS

**Implementation details.** For training the SCtrack model, two Nvidia A800 GPUs with 80G memory are adopted. We set the learning rate to $10^{-4}$ and decay it by a factor of $5$ every $40$ epochs. The total epoch number is 120 and the batch size is set to 32. Following the common training settings (Xu et al., 2023a; Nie et al., 2023), we construct template/search region inputs for two consecutive frames. Specifically, we randomly crop search regions that are located around the previous target within a predefined range. The detailed range hyper-parameters for different categories can be found in supplementary materials. We also adopt a simple data augmentation strategy to improve the tracking robustness. The random data augmentation simulates the change in target movement and heading angle in real tracking scenarios. A random translation along the $x$, $y$, and $z$ axes to targets in the current frame is applied to prevent the model from overfitting to the simple motions. Moreover, random horizontal flipping of points, along with their vertical axis within a certain range, is applied. The random translation and flipping are sampled from a Gaussian distribution with parameters. The hyperparameters can be found in supplementary materials.

**Datasets.** Following the widely-adopted settings (Hui et al., 2021; Xu et al., 2023a), our tracker is evaluated on three 3D SOT benchmarks, i.e., KITTI (Geiger et al., 2012), nuScenes (Caesar et al., 2019), and Waymo (Sun et al., 2020). The KITTI dataset consists of 21 point cloud sequences. We divide the sequences into three parts for different purposes: sequences 0-16 for training, 17-18 for validation, and 19-20 for testing, as per the guidelines in Giancola et al. (2019). The nuScenes dataset includes 700 sequences for training and 150 sequences for validation. We report the performance on the validation set for the nuScenes benchmark. As for the Waymo dataset, we utilized $1,121$ tracklets based on the previous SOT setting (Pang et al., 2021). The point cloud sequences

Table 1: The performance of different methods on the KITTI datasets. "Mean" denotes the average results of four categories.

| | Method | Success | | | | | Precision | | | | |
|---|---|---|---|---|---|---|---|---|---|---|---|
| | Category | Car | Pedestrian | Van | Cyclist | Mean | Car | Pedestrian | Van | Cyclist | Mean |
| | Frame Num. | 6424 | 6088 | 1248 | 308 | 14068 | 6424 | 6088 | 1248 | 308 | 14068 |
| KITTI | SC3D (Giancola et al., 2019) | 41.3 | 18.2 | 40.4 | 41.5 | 31.2 | 57.9 | 37.8 | 47.0 | 70.4 | 48.5 |
| | P2B (Qi et al., 2020) | 56.2 | 28.7 | 40.8 | 32.1 | 42.4 | 72.8 | 49.6 | 48.4 | 44.7 | 60.0 |
| | MLVSNet (Wang et al., 2021c) | 56.0 | 34.1 | 52.0 | 34.3 | 45.7 | 74.0 | 61.1 | 61.4 | 44.5 | 66.6 |
| | LTTR (Cui et al., 2021) | 65.0 | 33.2 | 35.8 | 66.2 | 48.7 | 77.1 | 56.8 | 45.6 | 89.9 | 65.8 |
| | BAT (Zheng et al., 2021) | 60.5 | 42.1 | 52.4 | 33.7 | 51.2 | 77.7 | 70.1 | 67.0 | 45.4 | 72.8 |
| | PTTR (Zhou et al., 2021) | 65.2 | 50.9 | 52.5 | 65.1 | 58.4 | 77.4 | 81.6 | 61.8 | 90.5 | 77.8 |
| | PTT (Shan et al., 2021) | 67.8 | 44.9 | 43.6 | 37.2 | 55.1 | 81.8 | 72.0 | 52.5 | 47.3 | 74.2 |
| | V2B (Hui et al., 2021) | 70.5 | 48.3 | 50.1 | 40.8 | 58.4 | 81.3 | 73.5 | 58.0 | 49.7 | 75.2 |
| | STNet (Hui et al., 2022) | 72.1 | 49.9 | 58.0 | 73.5 | 61.3 | 84.0 | 77.2 | 70.6 | 93.7 | 80.1 |
| | BAT (Zheng et al., 2021) | 60.5 | 42.1 | 52.4 | 33.7 | 51.2 | 77.7 | 70.1 | 67.0 | 45.4 | 72.8 |
| | GLT-T (Nie et al., 2023) | 68.2 | 52.4 | 52.6 | 68.9 | 60.1 | 82.1 | 78.8 | 62.9 | 92.1 | 79.3 |
| | CXTrack(Xu et al., 2023a) | 69.1 | 67.0 | 60.0 | 74.2 | 67.5 | 81.6 | 91.5 | 71.8 | 94.3 | 85.3 |
| | SyncTrack(Ma et al., 2023) | 73.3 | 54.7 | 60.3 | 73.1 | 64.1 | 85.0 | 80.5 | 70.0 | 93.8 | 81.9 |
| | MBPTrack (Xu et al., 2023b) | 73.4 | 68.6 | 61.3 | 76.7 | 70.3 | 84.8 | 93.9 | 72.7 | 94.3 | 87.9 |
| | SCtrack | **73.8** | **68.6** | **70.7** | **75.2** | **71.5** | **85.4** | **92.9** | **83.8** | **94.4** | **89.1** |

Table 2: The performance of different methods on the Waymo dataset. Each category is split into three difficulty levels: "Easy", "Medium", and "Hard". "Mean" denotes the average results of three levels. Note that except for our SCtrack, the results of other methods are obtained by running the official codes.

| | Method | Vehicle | | | | Pedestrian | | | | |
|---|---|---|---|---|---|---|---|---|---|---|
| | Split | Easy | Medium | Hard | Mean | Easy | Medium | Hard | Mean | Mean |
| | Frame Num. | 67832 | 61252 | 56647 | 185731 | 85280 | 82253 | 74219 | 241752 | 427483 |
| Success | P2B (Qi et al., 2020) | 57.1 | 52.0 | 47.9 | 52.6 | 18.1 | 17.8 | 17.7 | 17.9 | 33.0 |
| | BAT (Zheng et al., 2021) | 61.0 | 53.3 | 48.9 | 54.7 | 19.3 | 17.8 | 17.2 | 18.2 | 34.1 |
| | V2B (Hui et al., 2021) | 64.5 | 55.1 | 52.0 | 57.6 | 27.9 | 22.5 | 20.1 | 23.7 | 38.4 |
| | STNet (Hui et al., 2022) | 65.9 | 57.5 | 54.6 | 59.7 | 29.2 | 24.7 | 22.2 | 25.5 | 40.4 |
| | CXTrack (Xu et al., 2023a) | 63.9 | 54.2 | 52.1 | 57.1 | 35.4 | 29.7 | 26.3 | 30.7 | 42.2 |
| | MBPTrack (Xu et al., 2023b) | **68.5** | **58.4** | **57.6** | **61.9** | 37.5 | 33.0 | 30.0 | 33.7 | 46.0 |
| | SCtrack | 66.1 | 57.2 | 56.6 | 59.9 | **43.4** | **36.6** | **31.1** | **46.9** | **46.8** |
| Precision | P2B (Qi et al., 2020) | 65.4 | 60.7 | 58.5 | 61.7 | 30.8 | 30.0 | 29.3 | 30.1 | 43.8 |
| | BAT (Zheng et al., 2021) | 68.3 | 60.9 | 57.8 | 62.7 | 32.6 | 29.8 | 28.3 | 30.3 | 44.4 |
| | V2B (Hui et al., 2021) | 71.5 | 63.2 | 62.0 | 65.9 | 43.9 | 36.2 | 33.1 | 37.9 | 50.1 |
| | STNet (Hui et al., 2022) | 72.7 | 66.0 | 64.7 | 68.0 | 45.3 | 38.2 | 35.8 | 39.9 | 52.1 |
| | CXTrack (Xu et al., 2023a) | 71.1 | 62.7 | 63.7 | 66.1 | 55.3 | 47.9 | 44.4 | 49.4 | 56.7 |
| | MBPTrack (Xu et al., 2023b) | **77.1** | **68.1** | **69.7** | **71.9** | 57.0 | 51.9 | 48.8 | 52.7 | 61.0 |
| | SCtrack | 73.5 | 66.8 | 68.0 | 68.7 | **65.0** | **56.5** | **50.4** | **57.6** | **62.7** |

have been categorized into easy, medium, and hard subsets based on the number of points in the first frame of each sequence.

**Evaluation metrics.** Success and Precision originated from one pass evaluation (Kristan et al., 2016) are two main evaluation metrics for 3D single object tracking. Success is measured by the Intersection over Union (IoU) between predicted and ground truth 3D bounding boxes. Precision is determined by the area under the curve (AUC) for the distance between the centers of the two 3D boxes. Additional details can be found in the supplementary materials.

## 4.2 COMPARISON WITH STATE-OF-THE-ART TRACKERS

**Results on KITTI & Waymo.** We compare our SCtrack with many representative State-Of-The-Art (SOTA) trackers, including recent Siamese trackers like CXTtrack and MBPTrack. As shown in Tab. 1, SCtrack shows superior performance across various categories, achieving the best mean Success and Precision rates of $72.0\%$ and $89.1\%$, respectively. Specifically, SCtrack surpasses the previous SOTA method, i.e., MBPTrack, by $1.7\%$ with an even higher running speed. Our method has a much simpler model structure with superior performance, unlike the complex model design in the recent Siamese-based tracker, MBPTrack. This showcases the potential and effectiveness of our framework, which utilizes a joint appearance matching and motion modeling pipeline to accurately compute the target's relative motion between two consecutive frames.

In order to test the effectiveness of our proposed methods on different datasets, we evaluate the Car and Pedestrian models trained on the KITTI dataset using the Waymo dataset following common settings (Xu et al., 2023b). The results, presented in Tab. 2, show that our SCtrack achieves a leading position in both tracking robustness and accuracy, especially in the Pedestrian category. This implies that SCtrack has a strong ability to generalize the new data domain.

Table 3: The performance of different methods on the Nuscenes. "Mean" denotes the average results of four categories. Performance is arranged as "Success/Precision"

| Method | | | | | | | |
|---|---|---|---|---|---|---|---|
| Category | Car | Pedestrian | Truck | Trailer | Bus | Mean | Mean by Category |
| Frame Num. | 64159 | 33227 | 13587 | 3352 | 2953 | 117278 | |
| SC3D (Giancola et al., 2019) | 22.31/21.93 | 11.29/12.65 | 30.67/27.73 | 35.28/28.12 | 29.35/24.08 | 20.70/20.20 | 25.78/22.90 |
| P2B (Qi et al., 2020) | 38.81/43.18 | 28.39/52.24 | 42.95/41.59 | 48.96/40.05 | 32.95/27.41 | 36.48/45.08 | 38.41/40.90 |
| PTT (Shan et al., 2021) | 41.22/45.26 | 19.33/32.03 | 50.23/48.56 | 51.70/46.50 | 39.40/36.70 | 36.33/41.72 | 40.38/41.81 |
| BAT (Zheng et al., 2021) | 40.73/43.29 | 28.83/53.32 | 45.34/42.58 | 52.59/44.89 | 35.44/28.01 | 38.10/45.71 | 40.59/42.42 |
| PTTR (Zhou et al., 2021) | 51.89/58.61 | 29.90/45.09 | 45.30/44.74 | 45.87/38.36 | 43.14/37.74 | 44.50/52.07 | 43.22/44.91 |
| GLT-T (Nie et al., 2023) | 48.52/54.29 | 31.74/56.49 | 52.74/51.43 | 57.60/52.01 | 44.55/40.69 | 44.42/54.33 | 47.03/50.98 |
| MBPTrack (Xu et al., 2023b) | 62.47/70.41 | 45.32/74.03 | 62.18/63.31 | 65.14/61.33 | 55.41/51.76 | 57.48/69.88 | 58.10/64.19 |
| SCtrack | **65.21/73.14** | **46.52/75.30** | **65.10/66.20** | **70.62/66.91** | **59.45/576.80** | **59.97/72.25** | **61.60/67.89** |

Table 4: Ablation studies on our joint matching and motion modeling backbone on KITTI. "Concat" denotes channel concatenation for the template and search features. "S.S.Embed." and "Space-corr." denote spatially structured feature embedding and space correlation, while "Point-wise" denotes the features extracted by PointNet++. Performance is reported as Suc/Prec.

| #Num. | Feature | Space-corr. | Concat | Data.Aug | Car | Pedestrian | Van | Cyclist |
|---|---|---|---|---|---|---|---|---|
| ① | S.S.Embed. | ✗ | ✓ | ✓ | 69.5/79.4 | 60.3/85.8 | 64.5/79.6 | 75.1/95.4 |
| ② | S.S.Embed. | ✓ | ✗ | ✗ | 70.2/81.5 | 62.1/87.3 | 66.8/81.8 | 75.3/95.5 |
| ③ | S.S.Embed. | ✓ | ✗ | ✓ | 73.3/84.9 | 64.8/90.3 | 68.9/83.8 | 75.4/94.3 |
| ④ | Point-wise | ✗ | ✓ | ✓ | 65.7/77.6 | 52.3/81.5 | 63.7/77.8 | 75.8/94.8 |
| ⑤ | Point-wise | ✓ | ✗ | ✓ | 69.4/80.1 | 55.6/85.1 | 66.3/80.2 | 76.4/96.2 |

**Results on NuScenes.** We further conduct experiments on the challenging NuScenes dataset to validate the effectiveness of SCtrack. The following SOTA methods, which mostly follow the Siamese tracking and motion-centric paradigms, are compared to our method: SC3D (Giancola et al., 2019), P2B (Qi et al., 2020), PTT (Shan et al., 2021), BAT (Zheng et al., 2021), PTTR (Zhou et al., 2021), M2Track (Zheng et al., 2022), GLT-T (Nie et al., 2023) and MBPTrack (Xu et al., 2023b). Our SCtrack achieves the leading position among all the comparison methods across all categories, as shown in Tab. 3. It is worth noting that SCtrack performs significantly better than other models in the categories of Trailer and Bus, outperforming recent MBPTrack by 5.48% and 4.04% Success rate, respectively. These findings suggest that our proposed SCtrack framework can achieve outstanding results even with limited training data. The results also indicate that our SCtrack is highly generalizable to the new data domain and can handle complex scenarios and hard cases.

**Speed analysis.** SCtrack with default hyper-parameters runs at a high speed of 60 FPS on a single RTX3090 GPU. SCtrack's speed outperforms the recent SOTA Siamese tracker MBPtrack (50FPS) and the motion-centric tracker M2track (57FPS). This is mainly due to SCtrack's simple model pipeline, which can exploit both appearance and motion clues within a single backbone.

## 4.3 ABLATION STUDIES

**Analysis of spatially structured representation.** To analyze the impact of the proposed spatially structured representation on tracking performance, we conduct an ablation study on SCtrack using the KITTI dataset. As shown in Tab. 4, the case (①) of using spatially structured embedding outperforms that of using point-wise embeddings (④) by a great margin in all four categories. Moreover, a similar phenomenon is observed even with direct concatenation between template and search frames. It fully validates that the spatially structured embedding is superior to the widely adopted point-wise feature embedding for point cloud tracking. This is attributed to the rich motion clues in spatially structured embedding space.

**Analysis of data augmentation.** To validate the effectiveness of our simple data augmentation strategy, we train the SCtrack model using the data augmentation introduced in M2Track, i.e., without Data. Aug in Tab. 4. Our SCtrack benefits greatly (64.5% to 79.6% in Van), indicating that the data augmentation strategy can effectively model complex motion patterns in point cloud scenarios.

**Analysis of space-correlated transformer.** We study the effects of the proposed space-correlated transformer (SCT) in Tab. 5 and Tab. 6. The architectural variants of SCT, including the embedding dimensions and layer numbers, have varying impacts on the tracker's efficiency and performance. In Tab. 5, we observe a significant speed decrease when the channel dimension and layer number are scaling to 256/512 and 2, respectively. However, the performance gain is relatively small (around

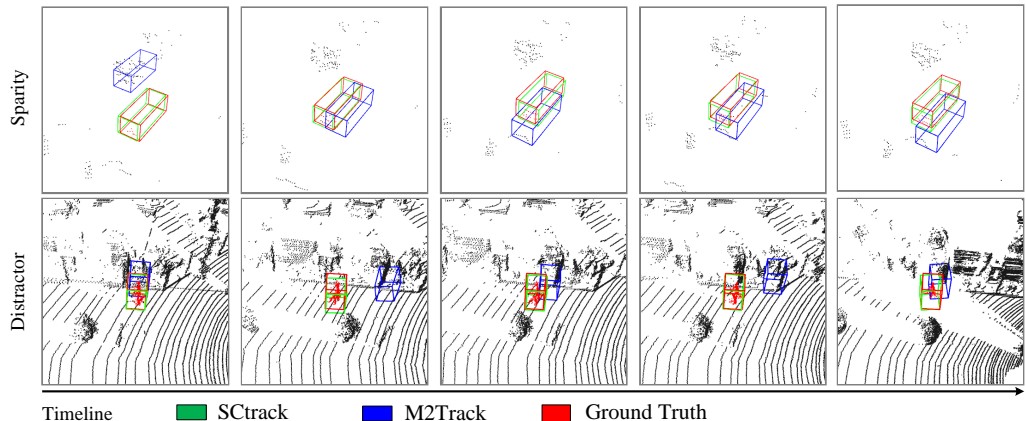

Figure 5: Visualized results on sparse and distracting point cloud scenarios (top and down row). It compares the motion-based tracker M2Track and our joint modeling method SCtrack.

Table 5: Architectural variants of the structured embedding module in the KITTI.

| #Num | Channle/Layer | Speed | Mean (Suc./Prec.) |
|---|---|---|---|
| ① | 128/1 | 61 FPS | 70.2/87.5 |
| ① | 128/2 | 45 FPS | 70.8/88.2 |
| ② | 256/1 | 40 FPS | 70.4/87.8 |
| ③ | 512/1 | 32 FPS | 70.1/87.6 |

Table 6: Analysis of region size for the space-correlated transformer in the KITTI.

| #Num | Region size | Mean (Suc./Prec.) |
|---|---|---|
| ① | fixed $3 \times 3$ | 70.1/87.3 |
| ② | fixed $5 \times 5$ | 70.8/88.1 |
| ③ | fixed $7 \times 7$ | 70.2/87.5 |
| ④ | learned | 71.5/88.9 |

$-0.1\%$ to $+0.6\%$). To better balance performance and speed, we set the channel dimension and layer number to $256$ and $1$, respectively. As shown in Tab. 6, our varied-size region scheme outperforms the best case $5 \times 5$ of the fixed region size by $0.7\%$ in terms of success, validating the effectiveness of adaptive region correlation.

**Analysis of robustness to sparsity and distractors.** Fig. 5 illustrates the superior performance of SCtrack over M2Track in sparse scenes, especially in extremely sparse ones. This is mainly due to the complementary effects of the appearance clues in our joint modeling framework, in contrast to solely exploiting motion clues. Our method also shows increased resilience to intra-class distractors.

## 5 LIMITATION AND DISCUSSION

Although SCtrack has proven the effectiveness of structured embedding in 3D SOT, its application to other Siamese-based and motion-based 3D trackers still needs to be validated and explored. Compared to the motion-centric tracking paradigm, our joint appearance matching and motion modeling backbone is conceptually simple and consists of two modules: a structured embedding module and a space-correlated transformer. However, our tracking model could be improved by unifying these two modules and embedding the space correlation operator into the structured embedding stage, resulting in a more straightforward model pipeline.

## 6 CONCLUSION

This paper introduces SCtrack, a conceptually simple tracking framework. SCtrack provides 3D single object tracking area with a joint exploring appearance matching and motion modeling paradigm. It explores the spatially structured representation of point clouds instead of widely used point-wise representations in most 3D SOT methods. Moreover, in contrast to the complex matching-based fusion module, SCtrack uses a space-correlated transformer module to capture the appearance and motion information of targets in varying sizes and shapes. Extensive experiments demonstrate our SCtrack framework is efficient, achieving superior performance over all previous state-of-the-art trackers. We expect this work could attract more attention to the balance between leveraging appearance and motion clues on 3D single object tracking task. A deeper exploration of the joint backbone may leave as future work.

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

# A  APPENDIX

## A.1  OVERVIEW OF APPENDIX

In the appendix, we first provide additional implementation details in Sec. B. Then, we present more experimental results and analysis in Sec. C. In the final section, we discuss our tracking framework in Section Sec. D.

# B  ADDITIONAL IMPLEMENTATION DETAILS

## B.1  STRUCTURED EMBEDDING MODULE.

We present the network details of the structured embedding module as follows: Our experimental setup is based on the LiDAR specifications of the KITTI dataset (Geiger et al., 2012). The detailed information are as follows: The range of point clouds for this task is set as $[-4.8, 4.8] \times [-4.8, 4.8] \times [-1.5, 1.5]$ meters along the $X$, $Y$, and $Z$ axes, respectively. Any points that go beyond the boundaries of the image will be removed. The voxel size is set to $v_D = 0.15, v_H = 0.075, v_W = 0.075$ meters. We have set T to 35, which represents the maximum number of randomly sampled points in each non-empty voxel. Our approach involves using two voxel encoding layers which is inspire by VoxelNet (Zhou & Tuzel, 2018): VE-1 (7, 32) and VE-2 (32, 128). The final channel dimension of the feature maps output to $\mathbb{R}^{128}$ by VE-2. We use Conv3D ($c_in$, $c_out$, k, s, p) to represent an 3D convolution operator. Here, $c_in$ and $c_out$ denote the number of input and output channels, and k, s, and p represent the kernel size, stride size, and padding size, respectively. The 3D convolutional filters in our voxel encoding layers are stacked as follows: Conv3D (128, 64, 3, (2,1,1), (1,1,1)); Conv3D (64, 64, 3, (1,1,1), (0,1,1)); Conv3D (64, 64, 3, (2,1,1), (1,1,1)) The input to the space-correlated transformer is a feature map of size $3 \times 384 \times 16 \times 16$ after reshaping. These dimensions correspond to the channel, height, and width of the 3D tensor.

## B.2  SPACE-CORRELATED TRANSFORMER.

In this section, we give details of the hyper-network and sampling process. The proposed hyper-network module differs from previous work (Liu et al., 2021) as it controls the regression distance without relying on a predefined parameter. Our space-correlated transformer follows the similar learned window scheme in VSA (Zhang et al., 2022). Specifically, we utilize the hypernetwork to predict the scales and offsets of the corresponding region, which are denoted as $s_w$ and $o_w$ in $R^2$. The normalization is achieved by multiplying the ratio between the window size and the image size, which are also referred to as $s_w$ and $o_w$ for simplicity. This normalization allows the hyper-network module to learn how to expand and move the current region toward the optimal attention region using the window as a reference. It then computes the tokens' new coordinates within that region.

### B.2.1  TEMPLATE AND SEARCH REGION INPUT.

In our method, we define search regions for two consecutive frames at timestamps $t - 1$ and $t$ by focusing on the previous target bounding box within a 3D space. Section 4.1 of the main manuscript outlines the specific range values of 3D spaces for different object categories. To ensure a fair comparison, we employ the same approach as existing methods (Qi et al., 2020; Zheng et al., 2022) by sampling 1024 points using farthest point sampling (Qi et al., 2017b) within the derived search regions as inputs for SCtrack. For SCtrack, instead of reducing the number of points in the point clouds, we partition the search regions into $128 \times 128 \times 20$ voxel grids, which are then used as inputs for the model.

### B.2.2  TRAINING DETAILS.

We train our proposed SCtrack models using the AdamW optimizer on two Tesla A800 GPUs, with a batch size of 32. In order to create the point cloud inputs for two consecutive frames at timestamps $t - 1$ and $t$, we crop search regions around the previous target within a specified range. For cars and humans categories, the ranges are set as $[\pm 4.8, \pm 1.5, \pm 1.92, \pm 1.5]$ to contain relevant points, respectively. Additionally, we apply random translations along the $xyz$ axes to the targets in the

---

**Algorithm 1** Training algorithm

---

```
def train(current_frame, prev_frame, prev_box):
    """
    Input: previous/current frame point cloud sequence; A target bounding box
        (x1, y1, z1, w1, l1, h1, theta_1)$ for the previous frame of the
        given sequence.
    """
    # Crop previous search region
    # Crop current search region
    # Extracting structured embedding from the point clouds
    # Send structured features into the space-correlated transformer
    # Obtain 4 Degree-of-Freedom relative motion
    """
    Output: a sequence of predicted bounding boxes (xt, yt, zt, wt, lt, ht,
        theta_t) from the second frame to the T frame
    """
    return output
```

---

current frame to simulate motion patterns and improve the model's accuracy. These translations are sampled from a Gaussian distribution with parameters $[\mu, \sigma]$. Based on the data we have, the target mostly moves along the $x$ axis. For the $x$, $y$, and $z$ axes, we set the mean $\mu$ and variance $\sigma$ to $[0, 0.3]$, $[0, 0.1]$, and $[0, 0.1]$, respectively.

### B.3 INFERENCE DETAILS.

During the inference stage, the models predict a target frame-by-frame in a continuous sequence of point clouds, given the target bounding box in the initial frame. Alg 1 presents the entire tracking process on a point cloud scene of SCtrack, respectively.

---

**Algorithm 2** Inference algorithm

---

```
def train(current_frame, prev_frame, prev_box):
    """
    Input: previous/current frame point cloud sequence; A target bounding box
        (x1, y1, z1, w1, l1, h1, theta_1)$ in the first frame of the given
        sequence.
    """
    # Crop previous search region
    # Crop current search region
    # Extracting structured embedding from the point clouds
    # Send structured features into the space-correlated transformer
    # Obtain 4 Degree-of-Freedom relative motion
      # Transform 4 relative motion with previous target bounding box
    """
    Output: a sequence of predicted bounding boxes (xt, yt, zt, wt, lt, ht,
        theta_t) from the second frame to the T frame
    """
    return output
```

---

## C ADDITIONAL EXPERIMENTS

### C.1 LEARNED REGION SIZE.

As shown in Fig. 7, compared to fixed hand-crafted regions, our SCT has the flexibility to determine the target region size. We set the default region size of our SCT in the main manuscript to be $4 \times 4$, which is roughly an average value for different target shapes. Fig. 7 shows that the learned region size tend to fit a uniform distribution for a variety of target shapes.

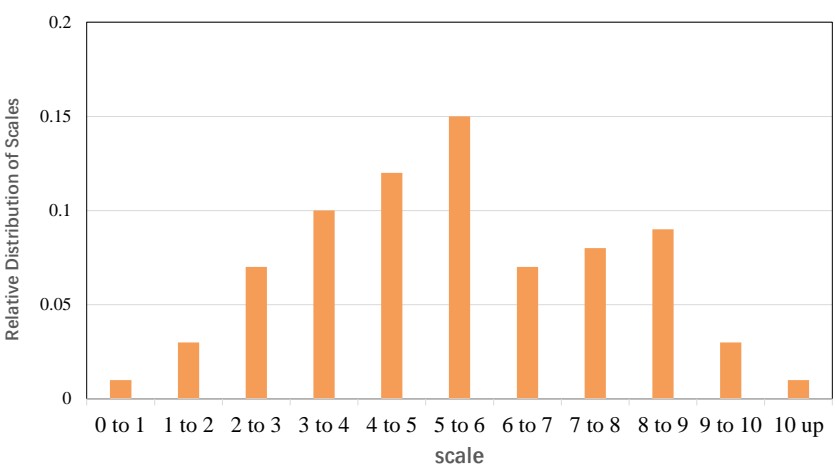

Figure 6: The relative distribution of scales is estimated by hyper-network. The X-axis denotes the scale of the varied-size region w.r.t. the default $4 \times 4$. The hyper-network generates the target regions at various scales to capture rich contextual information.

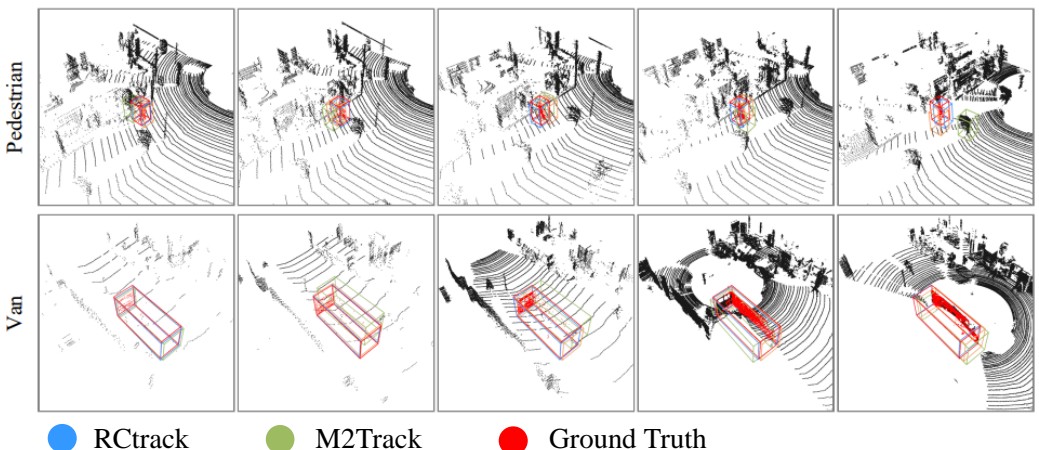

Figure 7: Visual results of our SCtrack and M2Track on the point cloud sequences of Pedestrian and Van categories (from top to bottom). The red points and box are the foreground points and ground truth of targets. The blue and green boxes denote the prediction results by our SCtrack and M2Track, respectively.

## D  ADDITIONAL DISCUSSIONS

### D.1  COMPUTATION COMPLEXITY ANALYSIS OF SCT.

The extra computations caused by SCT come from the hyper-network module, while the other parts, including the region-based multi-head self-attention and FFN network, remain the same as the transformer-based trackers. When given input features $X \in \mathcal{R}^{H \times W \times C}$, the hyper-network first uses a depth-wise convolutional layer with $7 \times 7$ kernels, and then an average pooling layer with a kernel size and stride equal to the region size. In the final, a convolutional layer with a kernel size of $1 \times 1$ is applied to predict the learned window size scale and offset movement. As we adopt the window-wise attention scheme, the computational complexity is not quadratic. We define the total window number as $N$ and the size of the window as $w$. Thus, the whole computational complexity of the hyper-network is computed as follows: $\mathcal{O}(49HWC)$, $\mathcal{O}(HWC)$, and $\mathcal{O}(\frac{N}{w^2}HWC)$. The computation complexity is linear with the number of input tokens. The inference latency and the cal-

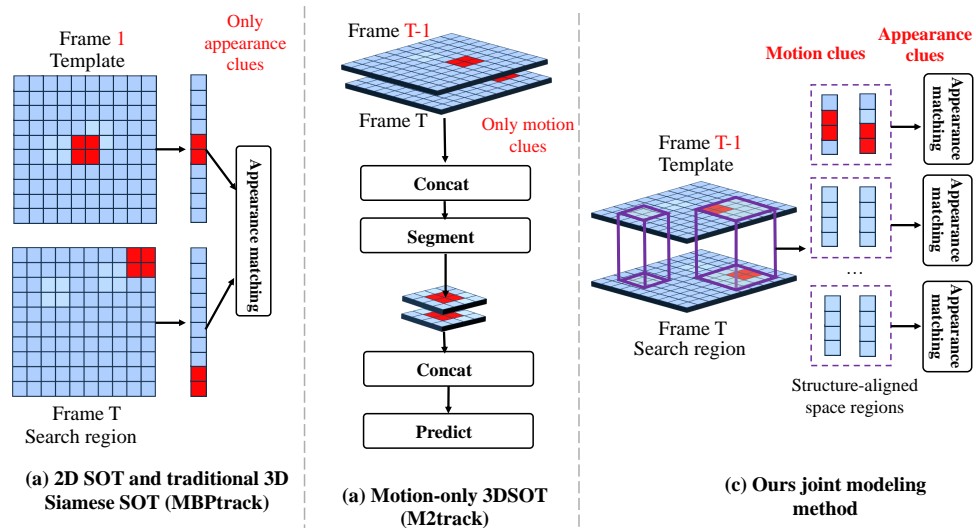

Figure 8: The illustration shows how appearance and motion clues are exploited in three different frameworks. Better zoomed in.

|  | SCT | Model Size | Flops | Speed |
|---|---|---|---|---|
| Siamese baseline | NA | 32.5M | 1.28G | 68FPS |
| SCtrack | ×1 | 35.4M | 1.41G | 61FPS |

Table 7: Increased model cost of our SCtrack.

culated Flops show that our model is far less (less than or equal to 5%) than the total computational cost of the baseline models.

## D.2 MOTION EXPLOITATION IN SCTRACK AND OTHER 3D SOT TRACKERS.

The difference in the inference phase between the 3D SOT trackers and the 2D SOT trackers (Hui et al., 2021; Xu et al., 2023a;b; Zheng et al., 2021; 2022) lies in the selection of the template. The appearance information in 2D images is far more rich than the 3D point cloud data. Thus, 2D SOT trackers generally take the first frame of the video as the template and adopt a fixed template strategy for subsequent video frames. In 3D SOT trackers, the predicted target location is center-cropped as the new template for next-frame prediction, which means the template is kept updating every time stamp. Thus, 3D SOT trackers only need to predict the relative motion of the tracked target between the current frame and the last frame. The reason is the point cloud data lacks enough appearance information, so exploiting motion clues is vital to locating the tracked target. Fig. 8 shows how we exploit the motion clues. The motion-centic M2track in 3DSOT leverages motion clues by using a direct spatial concatenation and a segmentation network to crop the aligned region. This complex operation can be simplified by our varied-size space region alignment. Our Method further conducts feature matching to exploit appearance clues as a complementary. Besides, our joint modeling method is under a conceptually simple and neat (one-stage) network.

## D.3 MODEL COST

As shown in Fig.8 and Tab. 7, our joint model pipeline can leverage both motion and appearance clues by slightly increasing the model complexity (+8.9%). Our method still maintains a conceptually simple pipeline (one-stage), which is advantageous to the Siamese trackers and motion-centric two-stage trackers.

