# OpenReview forum: "Space-Correlated Transformer: Jointly Explore the Matching and Motion Clues in 3D Single Object Tracking"
_ICLR.cc/2025/Conference — ICLR 2025 Conference Withdrawn Submission_

### Official Review · Reviewer_zudj · 2024-11-03

**Soundness:** 3
**Presentation:** 3
**Contribution:** 2
**Rating:** 5
**Confidence:** 4

**Summary:**

The manuscript introduces SCtrack, a novel framework for 3D single object tracking (SOT) in LiDAR point clouds. SCtrack aims to address the limitations of current 3D SOT methods by jointly exploring appearance matching and motion clues within a conceptually simple model. The authors propose a spatially structured representation and a Space-Correlated Transformer (SCT) to capture motion and appearance cues. The method achieves promising results on KITTI, NuScenes, and Waymo datasets, demonstrating state-of-the-art performance with a high frame rate of 60 FPS on a single RTX3090 GPU.

**Strengths:**

* The integration of appearance matching and motion modeling within a single framework is a significant contribution to the field of 3D SOT. SCtrack's ability to leverage both cues is a step forward from traditional Siamese and motion-centric paradigms.
* The paper reports impressive performance metrics on three major benchmarks, outperforming current state-of-the-art methods, which is a strong point in favor of the proposed method.
* The high frame rate of 60 FPS indicates that SCtrack is not only effective but also efficient, making it suitable for real-time applications such as autonomous driving.

**Weaknesses:**

* The paper fails to articulate the significance of SOT compared to multi-object tracking (MOT) in the context of autonomous driving. A clear discussion on the unique challenges and benefits of SOT is missing, which is critical for understanding the practical implications of the work.
* The NuScenes dataset includes seven categories, yet the paper only reports results for five. The omission of two categories is not justified, and it raises questions about the comprehensiveness of the evaluation. This gap undermines the credibility of the performance claims.

**Questions:**

The significance of SOT compared to multi-object tracking (MOT) is necessary.

---

### Official Review · Reviewer_X8rZ · 2024-11-03

**Soundness:** 3
**Presentation:** 3
**Contribution:** 3
**Rating:** 5
**Confidence:** 4

**Summary:**

The paper introduces SCtrack for 3D single object tracking (SOT), targeting limitations in existing methods, i.e., the limited appearance information in Siamese matching-based approaches and the complexity of motion-centric trackers. SCtrack combines appearance matching and motion cues within a single, efficient model by embedding point clouds into spatially structured features. It uses a space-correlated transformer to leverage spatially aligned region-based correlation, enhancing motion modeling and appearance matching. Moreover, to improve adaptability to various target shapes and sizes, a hyper-network module dynamically adjusts correlation region sizes. SCtrack achieves good performance on the KITTI, nuScenes, and Waymo datasets.

**Strengths:**

- Combining appearance matching and motion modeling in a unified framework is interesting.

- The structured voxel-based embedding and adaptive space-correlated transformer enhance its ability to capture motion patterns.

- SCtrack achieves good results on multiple datasets while maintaining a high speed of 60 FPS.

**Weaknesses:**

1. The authors could discuss and compare their method with recent 2024 works, such as:
- Xu, A., Nie, J., He, Z. and Lv, X., 2024. TM2B: Transformer-Based Motion-to-Box Network for 3D Single Object Tracking on Point Clouds. IEEE Robotics and Automation Letters.
- Sun, S., Wang, C., Liu, X., Shi, C., Ding, Y. and Xi, G., 2024. Spatio-Temporal Bi-directional Cross-frame Memory for Distractor Filtering Point Cloud Single Object Tracking. arXiv preprint arXiv:2403.15831.
- Luo, Z., Zhang, G., Zhou, C., Wu, Z., Tao, Q., Lu, L. and Lu, S., 2024, March. Modeling continuous motion for 3d point cloud object tracking. In Proceedings of the AAAI Conference on Artificial Intelligence (Vol. 38, No. 5, pp. 4026-4034).

2. In the abstract, the authors claim a high speed of 60 FPS; comparing inference speeds with other methods in the experiments section would improve clarity.

3. It would be better to include the final performance in Table 4 for better comparison.

4. The authors discuss data augmentation in the experiments and ablation studies but do not mention it in the introduction or methods. It would be beneficial to elaborate on this in the methods section if it is considered an important contribution.

**Questions:**

Given the claim of 60 FPS in the abstract, how does SCtrack's inference speed compare with that of other state-of-the-art methods under similar hardware conditions?

Since data augmentation is discussed in the experiments, can the authors elaborate on its implementation in the methods section to clarify its impact on SCtrack's performance?

---

### Official Review · Reviewer_9TqP · 2024-11-04

**Soundness:** 3
**Presentation:** 2
**Contribution:** 2
**Rating:** 6
**Confidence:** 4

**Summary:**

The paper introduces an innovative approach by combining two prominent paradigms in 3D single-object tracking (SOT): the Siamese-like and motion-centric frameworks. The authors propose a Spatially Structured Embedding to replace the conventional Siamese-like matching module and leverage a Space-Correlated Transformer to effectively capture both appearance and motion cues. This integration of methodologies offers a unique contribution to the 3D SOT domain.

**Strengths:**

1. The authors propose a Spatially Structured Embedding to replace the conventional Siamese-like matching module;

2. This work leverages a Space-Correlated Transformer to effectively capture both appearance and motion cues. This integration of methodologies offers a unique contribution to the 3D SOT domain.

**Weaknesses:**

1. Clarity in Figure 3: Figure 3 is somewhat confusing. Specifically, it is unclear how, after the squeeze operation, two tensors with shapes $C \times H \times W$, $C \times H \times W$ are transformed into a $2C \times D \times H \times W$ tensor. Further clarification on the origin and role of $D$ in this transformation would be beneficial.

2.  Figure 4 illustrates the modeling of motion cues, but the motivation for using varied sizes to enhance motion clue modeling is not sufficiently explained. Additionally, it remains unclear how these varied sizes are utilized in subsequent layers. It appears there may be a typo in *Equation (5)*, as the authors describe using varied-size regions, yet the equation employs ${VF}_{R}^{default}$. Further clarification on this point would improve the reader's understanding.

3. Explanation of Performance on Waymo Dataset: Although the authors achieve state-of-the-art performance on the KITTI and nuScenes datasets, SCTrack does not outperform MBPTrack on the Waymo dataset. The authors should provide an explanation for this discrepancy to offer insights into the model’s generalizability and performance variations across datasets.

4. Missing Citations: Several recent works that are relevant to the proposed approach appear to be missing citations:
    - (TPAMI2024) Exploring Point-BEV Fusion for 3D Point Cloud Object Tracking with Transformer

    Including these citations would strengthen the contextualization of the proposed work within the existing literature.

**Questions:**

Please refer to Weaknesses part.

---

### Official Review · Reviewer_hw1Y · 2024-11-04

**Soundness:** 2
**Presentation:** 2
**Contribution:** 2
**Rating:** 5
**Confidence:** 3

**Summary:**

This paper proposes a new 3D LiDAR Single Object Tracking (SOT) method named SCtrack. The pipeline consists of three parts: (1) feature extraction: the authors adopt points voxelization and 3D convolution to extract BEV features of template and search regions; (2) region correlation transformer: the BEV features of template and search regions are concatenated to build structured features VF. Then VF is divided into M regions. Each region will predict a scale and offset factor to transform fixed-size regions to varied-size space regions. Finally, an attention module is adopted to aggregate features from varied-size regions into queries generated from fixed-size regions. (3) Prediction Head: a max pooling layer followed by a linear layer is adopted to the updated VF to predict bounding box regression parameters. The authors conduct experiments on three 3D SOT benchmarks: KITTI, nuScenes, and Waymo. The proposed method achieves state-of-the-art performance on KITTI, nuScenes, and Pedestrian tracking on Waymo.

**Strengths:**

1. The proposed method is simple and straightforward to follow, and it achieves higher performance compared to recent state-of-the-art methods on several benchmarks.
2. The authors conduct ablation studies to validate the effectiveness of the proposed modules: (1) spatial structured embedding vs. point-wise; (2) varied-size region correlation transformer.

**Weaknesses:**

1. Using spatial BEV representation is very common in the field of 3D LiDAR detection. Using such representation in the SOT field can not been considered as a significant contribution. The authors need to explain in their rebuttal whether there are any previous works in SOT that have used similar representations.
2. In the presentation, the authors used many formulas and complex concepts to describe some commonly known representations. For example, voxel partition, sampling, and encoding are actually very common voxelization [1] processes in the 3D detection field. Additionally, the notations are somewhat complex and lengthy, such as in Equation 3 and Equation 5.
3. The authors should include the definition of default regions in the main text, otherwise readers cannot understand why the entire SOT pipeline lacks the involvement of 3D bounding box B_(i-1).

[1] End-to-End Multi-View Fusion for 3D Object Detection in LiDAR Point Clouds. CoRL 2019.

**Questions:**

1. The author should explain why the proposed method can not achieve higher performances comparing to MBPTrack in Vehicle class on Waymo dataset.
2. The authors need to improve their presentation style by using domain-specific concepts to explain the entire pipeline, which would make the method more comprehensible to readers.

---

### Official Review · Reviewer_6QG4 · 2024-11-06

**Soundness:** 3
**Presentation:** 2
**Contribution:** 2
**Rating:** 5
**Confidence:** 5

**Summary:**

This paper proposes SCtrack, which embeds point clouds into spatially structured features and conducts space correlation along the aligned spatial region. Specifically, the spatially structured representation in SCTrack inherently models motion clues among the consecutive frames of point clouds, thereby being complementary to appearance matching. A strategy of variedsize space regions is introduced to adapt to different target shapes and locations during space correlation. Experiments are conducted on mainstream benchmarks for evaluation including KITTI, NUSCENES and WAYMO.

**Strengths:**

- A space-correlated transformer tracker is proposed to combine appearance and motion matching into a unified backbone;
- The proposed approach achieves SOTA performance on large-scale benchmarks while maintaining high tracking speed;
- Sufficient experiments are conducted on KITTI, WayMO and NUSCENES;

**Weaknesses:**

- Incorrect comparison in Table 1: The "Mean" performance of the proposed SCTrack is calculated by averaging results across four categories. In contrast, other methods such as MBPTrack use sample-wise average results. This inconsistency leads to an unfair comparison. Please carefully verify and ensure consistency in all reported results in your paper.
- The paper proposes a joint matching and motion modeling tracking paradigm, which jointly explores appearance and matching information. However, this maybe a bit similar to the current Spatial-Temporal transformer based approaches like MBPTrack and StreamTrack [1]. StreamTrack performs both appearance and motion modeling via the Spatial-Temporal Relation matching, which can be regarded as a combination of previous appearance and motion paradigms; Please illustrate more differences with these approaches.
- Online tracking settings (i.e., evaluation) are unclear. During the online tracking stage, does SCTrack only uses the first frame template or maintain the updated templates?
- Some Minor issues: better to call ''SCTrack'' instead of ''SCtrack'', which looks more formally; Increase the spacing between the captions  in Table 5, Table 6 and Fig. 5 for improved readability.

[1] Modeling Continuous Motion for 3D Point Cloud Object Tracking;

**Questions:**

See Weakness. I would appreciate a rebuttal from the authors addressing the issues of incorrect comparisons, clarification of methodological differences, and an explanation of the tracking settings.

---

### Note · Authors · 2024-11-13

**Comment:**

The authors agree to withdraw this ICLR submission.

**Withdrawal Confirmation:**

I have read and agree with the venue's withdrawal policy on behalf of myself and my co-authors.